# Applications of machine learning in decision analysis for dose management for dofetilide

**Andrew E. Levy**[1], **Minakshi Biswas**[1], **Rachel Weber**[2], **Khaldoun Tarakji**[3], **Mina Chung**[3], **Peter A. Noseworthy**[4], **Christopher Newton-Cheh**[5], **Michael A. Rosenberg**[1,6]*

1 Division of Cardiology, University of Colorado Anschutz Medical Campus, Aurora, CO, United States of America, 2 Division of Biostatistics and Informatics, Colorado School of Public Health, Aurora, CO, United States of America, 3 Center for Atrial Fibrillation, Section of Cardiac Pacing and Electrophysiology, Cleveland Clinic Foundation, Cleveland, OH, United States of America, 4 Robert D. and Patricia E. Kern Center for the Science of Health Care Delivery, Mayo Clinic, Rochester, MN, United States of America, 5 Cardiovascular Research Center, Department of Medicine, Massachusetts General Hospital and Harvard Medical School, Boston, MA, United States of America, 6 Colorado Center for Personalized Medicine, University of Colorado Anschutz Medical Campus, Aurora, CO, United States of America

* michael.a.rosenberg@ucdenver.edu

## Abstract

### Background

Initiation of the antiarrhythmic medication dofetilide requires an FDA-mandated 3 days of telemetry monitoring due to heightened risk of toxicity within this time period. Although a recommended dose management algorithm for dofetilide exists, there is a range of real-world approaches to dosing the medication.

### Methods and results

In this multicenter investigation, clinical data from the Antiarrhythmic Drug Genetic (AAD-GEN) study was examined for 354 patients undergoing dofetilide initiation. Univariate logistic regression identified a starting dofetilide dose of 500 mcg (OR 5.0, 95%CI 2.5–10.0, p<0.001) and sinus rhythm at the start of dofetilide loading (OR 2.8, 95%CI 1.8–4.2, p<0.001) as strong positive predictors of successful loading. Any dose-adjustment during loading (OR 0.19, 95%CI 0.12–0.31, p<0.001) and a history coronary artery disease (OR 0.33, 95%CI 0.19–0.59, p<0.001) were strong negative predictors of successful dofetilide loading. Based on the observation that any dose adjustment was a significant negative predictor of successful initiation, we applied multiple supervised approaches to attempt to predict the dose adjustment decision, but none of these approaches identified dose adjustments better than a probabilistic guess. Principal component analysis and cluster analysis identified 8 clusters as a reasonable data reduction method. These 8 clusters were then used to define patient states in a tabular reinforcement learning model trained on 80% of dosing decisions. Testing of this model on the remaining 20% of dosing decisions revealed good accuracy of the reinforcement learning model, with only 16/410 (3.9%) instances of disagreement.

**Data Availability Statement:** Data cannot be shared publicly because of privacy concerns and violation of agreement of the informed consent

process. Data will be made available upon request from the Partners Healthcare Research Committee (Contact via jripton@partners.org), as well as the PI for AADGEN, Dr. Newton-Cheh (Contact via email at cnewtoncheh@mgh.harvard.edu), and corresponding author Dr. Rosenberg (Contact via email at michael.a.rosenberg@ucdenver.edu).

**Funding:** This work is supported by grants from the NIH T32 program (AEL: 5T32 HL007822) and the NIH NHLBI (MAR: 5K23 HL127296, CNC: R01 HL 143070). The funders had no role in study design, data collection and analysis, decision to publish, or preparation of the manuscript.

**Competing interests:** The authors have declared that no competing interests exist.

## Conclusions

Dose adjustments are a strong determinant of whether patients are able to successfully initiate dofetilide. A reinforcement learning algorithm informed by unsupervised learning was able to predict dosing decisions with 96.1% accuracy. Future studies will apply this algorithm prospectively as a data-driven decision aid.

## Background

Decision analysis is an emerging field that uses outcomes from different decision approaches to guide future decision-making[1]. In many cases, medical decisions can be formulated as Markov-decision processes (MDPs), in which a given state of conditions can predict future states based on a model for decision-making[2]. Reinforcement learning, a subset of machine learning (ML), expands on MDPs by embedding reward-based feedback into decision outcomes so that an optimal decision approach, termed the policy, can be identified[3]. In recent years, this approach has achieved supra-human success rates in video and board games, among other applications[4, 5].

Reinforcement learning is one of three main categories of ML gaining popularity in medical applications, the other two being supervised and unsupervised learning[6]. Supervised applications use an example dataset to learn general rules (an algorithm) about the relationship of predictor variables (termed "features") to an outcome of interest (termed a "label"). These general rules can then be applied to a new dataset to predict outcomes. Unsupervised learning, in contrast, does not use labelled outcomes and, instead, discovers relationships between different features on its own. The discovery process often restructures data into new classes, "shrinking" and consolidating features for more nimble use in supervised applications. In many applications, these methods complement each other, but whereas supervised and unsupervised methods lead to *descriptive* analyses, *feedback* from outcomes allows reinforcement learning to produce *prescriptive* analyses[7]. For this reason, reinforcement learning holds great promise as a tool to enrich clinical decisions. Currently, however, there are relatively few published applications in healthcare[8, 9].

Dofetilide is a common antiarrhythmic medication primarily used to treat atrial fibrillation. It is one of the few anti-arrhythmic medications other than amiodarone that has been approved for use in patients with coronary artery disease or cardiomyopathy. Like many other Vaughan Williams class III agents, dofetilide blocks the rapid delayed rectifier, $I_{Kr}$ current, and thus can cause QT prolongation. Due to the risk of resultant fatal arrhythmias, the FDA has mandated a 3-day monitoring period for drug initiation[10]. There is a recommended algorithm for making dose adjustments during initiation, but these adjustments are still made at the treating provider's discretion[10, 11]. In this investigation, we examine the patterns of dofetilide dose adjustment and the role of machine learning to develop algorithms aimed at successful initiation of the medication.

## Methods

This study has been approved by the University of Colorado Internal Review Board (COMIRB Protocol #16–2675), and the Partners Human Research Committee (#2013-P002623). All subjects provided written informed consent.

## Study population

The Antiarrhythmic Drug Genetic (AADGEN) study is a multi-center collaboration that includes investigators from the Massachusetts General Hospital (MGH, Boston, MA), Beth Israel Deaconess Medical Center (Boston, MA), the Boston-area Veterans Affairs Medical Center (West Roxbury, MA), the Cleveland Clinic (Cleveland, OH), the Mayo Clinic (Rochester, MN), and the University of Colorado Hospital (Aurora, CO). Patients were enrolled from July 7, 2014 to September 19, 2018, with the inclusion criterion being any patient admitted to in-patient telemetry for monitoring of initiation of dofetilide. The exclusion criteria included failure to provide written informed consent and failure to obtain a pre-dofetilide ECG. Massachusetts General Hospital served as the study's coordinating center for this investigation. Internal Review Board approval was obtained at all enrolling centers. This study is a sub-study of a larger investigation into the genetic predictors of cardiac repolarization and drug toxicity of antiarrhythmic medications (Clinicaltrials.gov identifier: NCT02439658).

Demographic and clinical information were obtained on all study participants that included age, height, weight, body mass index (BMI), medications, past medical and cardiac history, including history of pacemaker/defibrillator, atrial fibrillation, ventricular fibrillation, left ventricular function from transthoracic echocardiogram, recent lab values including creatinine, potassium, and magnesium, and electrocardiograms that include underlying rhythm, rate, and relevant intervals (PR, QRS, QT). QT interval was corrected for heart rate using Fridericia's formula[12]. The timing of electrical cardioversion was also recorded.

The outcome of interest was successful loading of dofetilide, defined as discharge on dofetilide at any dose after at least 5 administrations. Data for all participants was collected retrospectively, after completion of the hospitalization; no clinical adjustments or changes were made by treating physicians as part of this investigation. Data was maintained in a centralized RedCap database managed by the study coordinating center at MGH.

## Data processing

Prior to analysis, quality control was performed by study investigators, with manual review of outlier values for ECG parameters (i.e., QTc > 600 ms) and for discordant data values (e.g., PR interval on an ECG with rhythm listed as 'atrial fibrillation'). When resolution or validation was not possible, values were replaced as missing. Summary and descriptive statistics are based on analysis of non-missing data; only 4.2% of the total dataset was missing. Due to the restrictions of machine-learning algorithms for complete datasets, missing values needed to be imputed with the median for numerical and integer values and most common for categorical. Categorical variables were also coded using 'one-hot' encoding and numerical variables were rescaled using min-max rescaling. Dose adjustments were only included if they were a decrease in dose from a higher dose, as FDA guidelines for dofetilide initiation suggest starting at the highest dose based on kidney function, and adjusting downward based on the QT changes on ECG; as such, any dose increase during the hospitalization was off-label. Based on this criterion, 14 patients who underwent dose increases were excluded. For all model evaluations, data were split into training (80% of total data) and testing sets (20% of total data) at the patient level.

## Supervised analysis

Basic stepwise logistic regression was performed for successful initiation of dofetilide using a p value for exclusion of greater than 0.05. Based on the observation that dose adjustments were a significant predictor of successful initiation, we used ensemble methods to develop predictive models of the dose adjustment process. These models included L1 regularized logistic

regression, random forest classification, a boosted decision tree classifier, support vector classification (radial basis function kernel), and K-nearest neighbors classification with a maximum of 10 neighbors. Comparison measures included accuracy, precision and recall scores, $F_1$-score[13, 14], and area under ROC curve.

## Unsupervised analysis

For unsupervised analysis, we first performed principal component analysis. Plotting the number of principal components (PC) versus variance, we hoped to identify the number of PCs that would account for greater than 90% of the variability in the data. We then performed a cluster analysis based on within cluster variation (sum-of-squares), and used the 'elbow' method to determine cluster numbers with sufficiently low within-cluster variability. We then used a K-means approach to create these clusters for use in subsequent reinforcement learning analyses.

## Reinforcement learning

We next applied reinforcement learning using the SARSA algorithm (state–action–reward–state–action) for selecting dose adjustments based on a negative reward for unsuccessful initiation[15]. We applied two broad approaches to creation of action-value estimates (i.e., Q values) [16]. First, we defined 8 states created using K-means clustering from all clinical features, and performed tabular updates to a Q table based on dynamic programming (step-by-step updates). Alternatively, we performed linear function approximation for the Q values using linear weights (termed 'Q learning'[17]), with updates using stochastic gradient descent based on experience[15]. The available actions in the Q value estimates included 'continue the same dose' or 'decrease the dose'. The reward was selected to be -10 for doses leading to stopping of the medication (last dose before stopping) and 0 for all other doses, in order to penalize decisions resulting in a negative outcome.

The SARSA algorithm[15] updates a Q table with expected reward values based on state and action selected based on the following variation of the Bellman equation[15]:

$$Q_{new}(S_t,\ A_t) = Q_{old}(S_t,\ A_t) + \alpha^*[(R_t + \gamma^*Q(S_{t+1},\ A_{t+1})) - Q_{old}(S_t,\ A_t)]$$

The Q table was initialized at 0 for all values, with gamma (discount factor) of different values ranging from 0.1 to 1.0, and alpha (learning rate) of 0.1. Of note, a gamma close to 1 puts more weight on future states and rewards while a gamma of close to 0 tends to put more weight on immediate rewards. We experimented with a range of learning rates (0.05 to 0.3). The learning rate is the extent to which Q-values are updated with new iterations of data. Reinforcement learning algorithms were fitted with the testing set (per above, 80% of doses) and compared with actual decisions on the held-out test set (per above, 20% of doses). Additional analyses were performed using k = 4 and k = 6 (number of clusters).

## Analysis

Descriptive statistical analysis, including chi-square for categorical and t-test for continuous comparison, as well as univariate logistic regression, was performed using Stata IC, Version 15.1 (StataCorp, LLC, College Station, TX). Machine learning, including unsupervised, supervised, and reinforcement learning algorithms, were performed using Python 3, running scripts on Jupyter notebook (v5.0.0) deployed via Anaconda Navigator, on a Macbook Pro laptop computer (High Sierra, v10.13.6). Primary source of machine learning packages was *scikit-learn* (see *Supplemental Methods* for details).

**Table 1. Baseline demographics and clinical characteristics.** A total of 354 subjects were enrolled in the Anti-arrhythmic Drug Genetic (AADGEN) study, with successful initiation (discharged on dofetilide) in 310 patients (87.1%) and unsuccessful in 44. Note: Dose excludes 4 patients with a different starting dose than listed.

| | | Successful initiation (N = 310) | Unsuccessful initiation (N = 44) | P value |
|---|---|---|---|---|
| Age (Mean ± SD) | | 66.6 ± 10.7 | 67.7 ± 9.7 | 0.53 |
| Female Sex (%) | | 91 (29.4%) | 18 (40.9%) | 0.12 |
| BMI (Mean ± SD) | | 30.2 ± 7.2 | 29.6 ± 7.5 | 0.57 |
| History of: | | | | |
| | AF (%) | 297 (95.8%) | 44 (100%) | 0.17 |
| | VT (%) | 12 (3.9%) | 0 (0%) | 0.18 |
| | PPM (%) | 20 (6.5%) | 3 (6.8%) | 0.93 |
| | ICD (%) | 20 (6.5%) | 3 (6.8%) | 0.93 |
| | HTN (%) | 142 (45.8%) | 18 (40.9%) | 0.54 |
| | DM (%) | 38 (12.2%) | 3 (6.8%) | 0.29 |
| | CAD (%) | 68 (21.9%) | 7 (15.9%) | 0.36 |
| | CHF (%) | 35 (11.3%) | 8 (18.2%) | 0.19 |
| LV EF (%) (Mean ± SD) | | 54.8 ± 12.3 | 50.9 ± 16.2 | 0.10 |
| Medications: | | | | |
| | Beta blockers (%) | 117 (57.1%) | 27 (61.4%) | 0.59 |
| | Calcium channel blockers (%) | 67 (21.6%) | 17 (38.6%) | 0.01 |
| Baseline lab values: | | | | |
| | Potassium (mmol/L) | 4.3 ± 0.47 | 4.4 ± 0.36 | 0.28 |
| | Magnesium (mg/dL) | 2.0 ± 0.26 | 2.0 ± 0.19 | 0.98 |
| | Creatinine (mg/dL) | 1.01 ± 0.25 | 1.04 ± 0.28 | 0.46 |
| Baseline ECG: | | | | |
| | Sinus Rhythm (%) | 114 (37.8%) | 12 (27.3%) | 0.18 |
| | HR | 80.8 ± 20.5 | 86.3 ± 24.0 | 0.11 |
| | PR | 179.2 ± 40.8 | 190.2 ± 56.9 | 0.39 |
| | QRS | 102.4 ± 25.8 | 98.8 ± 25.1 | 0.38 |
| | QT | 428.2 ± 50.4 | 436.5 ± 59.4 | 0.33 |
| | QTc | 445.0 ± 39.2 | 451.9 ± 39.2 | 0.25 |
| Initial Dose | | | | |
| | 500 mcg | 227 (73.5%) | 25 (56.8%) | 0.02 |
| | 250 mcg | 74 (24.0%) | 16 (36.4%) | - |
| | 125 mcg | 4 (1.3%) | 3 (6.8%) | - |

Abbreviations: SD = Standard deviation; BMI = Body mass index; AF = Atrial fibrillation; VT = Ventricular tachycardia; PPM = Presence of a permanent pacemaker; ICD = Presence of an implantable cardioverter-defibrillator; HTN = Hypertension; DM = Diabetes; CAD = Coronary artery disease; CHF = Congestive heart failure; LV EF = Left ventricular ejection fraction; ECG = electrocardiogram; HR = heart rate; PR/QRS/QT = cardiac intervals (not abbreviations); QTc = corrected (heart rate) QT interval; mcg = micrograms.

## Results

The baseline characteristics of the cohort are shown in Table 1. A total of 354 subjects were enrolled, with successful initiation (discharged on dofetilide) in 310 patients (87.1%) and unsuccessful in 44. Use of calcium channel blockers and initial dose of dofetilide were different between patients with successful vs. unsuccessful initiation of dofetilide, although none of these p values reached statistical significance after Bonferroni adjustment for multiple comparisons (probability of false positive = p/(# of rows in Table 1) = 0.05/24 = 0.002). There were no other differences in baseline parameters between patients.

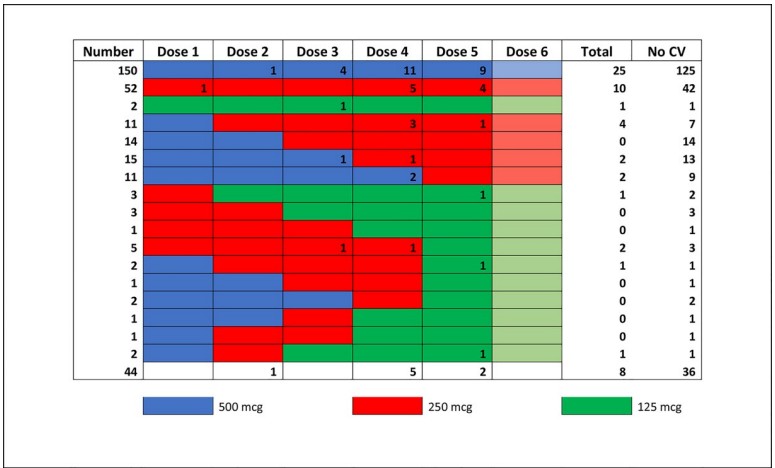

**Fig 1. Dose patterns of dofetilide.** A schematic of the most common dosing approaches for dofetilide (color-coded rows) among patients who were successfully initiated (discharged on medicine0. The numbers in each individual cell correspond to the number of electrical cardioversion procedures performed after that specific dose within that specific dosing scheme. 29 patients with atypical dosing regimens (i.e. increases in dose) are excluded. The bottom row represents patients who were not successfully initiated on Dofetilide (n = 44).

Fig 1 shows representative dosing approaches for dofetilide, as well as timing of cardioversions. The most common dose regimens included subjects with no adjustments throughout the 5–6 dose course in order to obtain a steady-state of the medication (n = 204, 57.6%). Stepwise univariate regression was performed for successful initiation across the course of dofetilide initiation, which revealed that dose number, dose amount, dose adjustment, ejection fraction, history of heart failure, sinus rhythm, QRS, QTc, presence of a pacemaker, and coronary artery disease were predictors of successful discharge on dofetilide at $p < 0.05$ (Table 2). The strongest predictors for successful initiation of dofetilide were starting dose of 500 mcg (OR 5.0, 2.5–10.0, $p < 0.001$) and dose adjustment during initiation (OR 0.19, 0.21–0.31, $p < 0.001$), which was a negative predictor. Because it had such a strong effect, we selected dose adjustment as the target for machine learning techniques.

**Table 2. Association with successful loading of dofetilide.** Univariate logistic regression results for associations with successful loading of dofetilide (discharged on medication). Dose position refers to an integer from 1 to 6, in which 1 would have been the first dose and 5 or 6 would have been the final dose. Dose adjustment is any decrease in dose from the prior dose. Sinus rhythm refers to patients in sinus rhythm at the time of the dosing decision.

| | OR | CI | p value |
|---|---|---|---|
| **500mcg dose**[*] | 5.0 | 2.5–10.0 | <0.001 |
| **250 mcg dose**[*] | 1.5 | 0.8–2.9 | 0.21 |
| **Dose position** | 1.3 | 1.1–1.5 | 0.001 |
| **Dose adjustment** | 0.19 | 0.12–0.31 | < 0.001 |
| **Sinus rhythm** | 2.8 | 1.8–4.2 | < 0.001 |
| **PPM** | 3.3 | 1.4–7.4 | 0.004 |
| **LVEF** | 1.03 | 1.01–1.05 | 0.001 |
| **CHF** | 1.8 | 1.0–3.0 | 0.04 |
| **QRS** | 1.02 | 1.01–1.03 | 0.001 |
| **QTc** | 0.992 | 0.987–0.997 | 0.002 |
| **CAD** | 0.33 | 0.19–0.59 | < 0.001 |

PPM = Presence of a pacemaker; LVEF = Left ventricular ejection fraction (by transthoracic echocardiogram); CHF = Congestive Heart Failure; QRS = QRS interval; QTc = Corrected QT interval; CAD = Coronary artery disease. [*]Comparison is with 125mcg dose.

**Table 3. Supervised learning approaches to decision-making.** A naïve approach to dose adjustment classification, in which dose adjustments were predicted based purely on the basis of a dose change probability of 7.1%, was used as a comparator for supervised approaches to predict dose adjustments.

| | Accuracy | Precision Score | Recall Score | F1 Score | AUC |
|---|---|---|---|---|---|
| Naïve (Probabilistic) Classifier | 0.93 | 0.0 | 0.0 | 0.0 | 0.5 |
| L1 Logistic Regression | 0.93 | 0.0 | 0.0 | 0.0 | 0.5 |
| Random Forest Classifier | 0.93 | 0.0 | 0.0 | 0.0 | 0.5 |
| Boosted Decision Tree | 0.93 | 0.5 | 0.03 | 0.065 | 0.52 |
| SVM with RBF kernel | 0.93 | 0.0 | 0.0 | 0.0 | 0.5 |
| KNN (k = 1) | 0.86 | 0.14 | 0.17 | 0.15 | 0.54 |
| KNN (k = 10) | 0.93 | 0.0 | 0.0 | 0.0 | 0.5 |

SVM = Support vector machine, RBF = Radial basis function, KNN = K-nearest neighbor classification, Accuracy = # correct/total; precision score (positive predictive value) = # of true positives/(true positives + false positives); recall score (sensitivity) = # of true positives/(true positives + false negatives); F1 score = 2 * (precision*recall)/(precision + recall); AUC = area under receiver operator characteristic curve.

The 354 subjects in our analysis collectively received a total of 2037 doses of dofetilide. Out of a possible 2037 opportunities to adjust the dose of dofetilide, dose adjustments were made in 144 instances. This corresponds to a dose change probability of 7.1%, indicating that a naïve approach that predicted only no dose adjustment would be accurate 92.9% of the time, which was used as the comparison for machine-learning approaches developed to predict whether a dose adjustment would be made. However, none of the supervised analyses resulted in improvement in identification of a medication adjustment by providers over a naïve approach (based on accuracy, or any of the other classification metrics applied) as shown in Table 3.

As described above, unsupervised principal component analysis was performed across 25 patient and dosing characteristics. We noted that the first two principal components (PCs) accounted for 65.0% of the total variance and 90% of the total variance could be explained by the first 8 PCs (Fig 2A). Cluster analysis using within-cluster sum-of-squares identified cluster numbers of k = 4 or greater as providing sufficiently low within-cluster variability, and validated use of k = 8 clusters (Fig 2B). Qualitative assessment of each PC revealed that there was apparent clustering along the first PC into 6 groups, which likely represent the dose number (S1 Fig). Characteristics of each PC cluster are described in Table 4.

After training the model on the training set (80% of data, 1627 doses), the accuracy of a tabular reinforcement-learning model for predicting actual decisions on the testing set (20%, 410 doses) was good, with only 3.9% disagreement (16/410) noted. Sensitivity analysis using a range of learning rates (alpha) and discount rates (gamma) had no impact on the accuracy of prediction; only the absolute Q values changed (not relative values). The least disagreement was observed in the Q table cluster with the smallest (most negative) values for rewards (Table 5). The analysis was repeated with use of k = 4 (S1 Table) and k = 6 clusters (S2 Table) which predicted actual decisions with less accuracy than the model with k = 8 clusters (98/410, 23%, correct for k = 4 clusters and 336/410, 82%, correct for k = 6 clusters).

A linear reinforcement-learning policy function was able to achieve equal accuracy to tabular learning for certain hyper-parameter choices (alpha and gamma). Unlike the tabular learning model, however, the linear model was highly labile depending on hyper-parameter choices (S2 Fig). These models also had unstable weight estimates (See S3 Table) across parameters.

## Discussion

In this investigation of decision-making surrounding dofetilide initiation, we examined several approaches for evaluating dose adjustment decisions. It is important to note that while

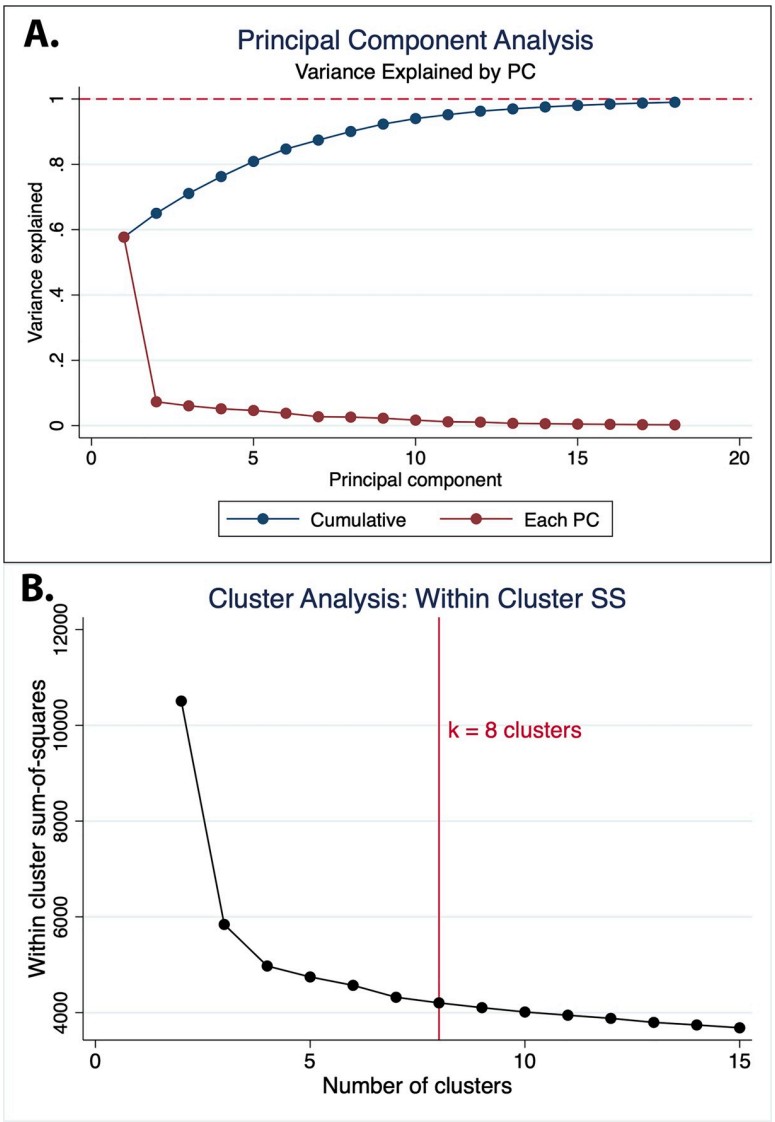

**Fig 2. Principal component analysis. A.** Cumulative and per-component variance explained for each sequential principal component (PC). **B.** Cluster analysis based on within-cluster sum-of-squares.

dofetilide initiation is performed in the hospital primarily for safety reasons (adverse event monitoring), the goal of these admissions is successful initiation of the drug (discharge on dofetilide) while minimizing the risk of subsequent TdP or potentially fatal ventricular arrhythmias[11]. With this in mind, there are important insights to be drawn from this novel application of advanced analytics and machine learning to decision-making surrounding dofetilide initiation.

First, it was evident from several models that making dose adjustments, particularly at later time points, was associated with less probability of successful initiation of the medication. This association was evident in both simple logistic regression models, as well as reinforcement-learning models in which the cluster with the most negative reward (#5) was composed of doses at a later state in the hospitalization (dose 4–5 vs. 1–2), and of smaller size. This finding suggests that making a decision to lower the dose of dofetilide in a patient who has already received 3–4 doses and is already on a lower dose (250 or 125mcg) is very unlikely to result in

**Table 4. Cluster characteristics.** Unsupervised principal component analysis was performed across 25 patient and dosing characteristics.

| Cluster | 1 | 2 | 3 | 4 | 5 | 6 | 7 | 8 |
|---|---|---|---|---|---|---|---|---|
| **Number** | 241 | 229 | 255 | 287 | 369 | 251 | 184 | 221 |
| **Dose position (% of dose)** | *2*–125 (51.9%) *3*–116 (48.1%) | *4*–229 (100%) | *1*–255 (100%) | *5*–166 (57.9%) *6*–121 (42.2%) | *5*–167 (45.3%) *6*–202 (54.7%) | *3*–135 (53.8%) *4*–116 (46.2%) | *2*–98 (53.3%) *3*–86 (46.7%) | *1*–99 (44.8%) *2*–122 (55.2%) |
| **Dose amount** | *500mcg*-241 (100%) | *500mcg*-182 (82.0%) *250mcg*-40 (18.0%) | *500mcg*-255 (100%) | *500mcg*-287 (100%) | *250mcg*-218 (79.9%) *125mcg*-55 (20.1%) | *250mcg*-197 (84.6%) *125mcg*-36 (15.5%) | *500mcg*-184 (100%) | *250mcg*-188 (90.8%) *125mcg*-19 (9.2%) |
| **Age (years)** | 62.6 ± 10.5 | 64.6 ± 10.8 | 64.6 ± 10.2 | 64.9 ± 9.8 | 68.0 ± 10.9 | 70.1 ± 10.2 | 67.3 ± 8.3 | 70.8 ± 10.6 |
| **Female Sex** | 55 (22.8%) | 48 (21.0%) | 64 (25.1%) | 61 (21.3%) | 138 (37.4%) | 113 (45.0%) | 44 (23.9%) | 99 (44.8%) |
| **Sinus Rhythm** | 125 (52.3%) | 158 (70.5%) | 93 (37.1%) | 229 (80.6%) | 284 (79.8%) | 125 (51.4%) | 87 (48.1%) | 86 (40.4%) |
| **Heart rate (bpm)** | 74.7 ± 17.0 | 68.2 ± 15.4 | 80.7 ± 20.1 | 65.9 ± 13.6 | 70.0 ± 17.6 | 73.6 ± 18.5 | 71.9 ± 16.5 | 78.5 ± 21.0 |
| **QRS** | 100.0±21.2 | 103.7±24.1 | 102.8±24.9 | 104.3±24.8 | 100.9±24.0 | 102.5±30.9 | 107.5±38.3 | 103.3±26.5 |
| **QTc** | 465.1±34.5 | 469.5±35.1 | 443.7±35.6 | 468.6±35.2 | 477.1±39.0 | 486.1±42.2 | 463.1±36.6 | 466.6±46.7 |
| **Creatinine** | 0.96±0.21 | 1.00±0.25 | 0.98±0.22 | 0.98±0.23 | 1.04±0.27 | 1.07±0.28 | 0.99±0.22 | 1.09±0.31 |
| **Beta Blocker** | 122 (50.6%) | 113 (49.3%) | 144 (56.5%) | 162 (56.5%) | 217 (58.8%) | 173 (68.9%) | 108 (58.7%) | 138 (62.4%) |
| **CCB** | 39 (16.2%) | 54 (23.6%) | 53 (20.8%) | 59 (20.6%) | 90 (24.4%) | 57 (22.7%) | 59 (32.1%) | 61 (27.6%) |
| **CHF** | 12 (5.0%) | 17 (7.4%) | 27 (10.6%) | 24 (8.4%) | 54 (14.6%) | 47 (18.7%) | 26 (14.1%) | 39 (17.7%) |
| **CAD** | 24 (10.0%) | 31 (13.5%) | 47 (18.4%) | 47 (16.4%) | 88 (23.9%) | 79 (31.5%) | 49 (26.6%) | 58 (26.2%) |
| **HTN** | 0 (0%) | 81 (35.4%) | 106 (41.6%) | 121 (42.2%) | 182 (49.3%) | 139 (55.4%) | 184 (100%) | 110 (49.8%) |
| **DM** | 12 (5.0%) | 22 (9.6%) | 31 (12.2%) | 35 (12.2%) | 42 (11.4%) | 33 (13.2%) | 41 (22.3%) | 23 (10.4%) |
| **PPM** | 14 (5.8%) | 14 (6.1%) | 15 (5.9%) | 20 (7.0%) | 25 (6.8%) | 16 (6.4%) | 14 (7.6%) | 13 (5.9%) |
| **ICD** | 11 (4.6%) | 12 (5.2%) | 16 (6.3%) | 16 (5.6%) | 22 (6.0%) | 22 (8.8%) | 11 (6.0%) | 18 (8.1%) |
| **LVEF** | 54.6 ± 12.6 | 54.7 ± 12.3 | 54.3 ± 13.0 | 54.4 ± 13.0 | 54.5 ± 12.1 | 53.9 ± 13.0 | 53.6 ± 13.0 | 54.3 ± 13.1 |

All values listed at mean ± SD or number (%). Sinus rhythm = sinus or atrial paced rhythm (not atrial fibrillation/flutter); CCB = Calcium channel blocker; CHF = heart failure; CAD = coronary artery disease; HTN = hypertension; DM = diabetes mellitus; PPM = pacemaker present; ICD = implantable cardioverter-defibrillator present; LVEF = left ventricular ejection fraction based on transthoracic echocardiography

successful initiation. While further work is needed to validate these models prospectively, this finding could have an important impact on reducing healthcare costs. It would save time and money to stop the initiation process early in a patient in whom the probability of successful initiation is unlikely, rather than staying another day or night in the hospital, or perhaps start at a lower dose in patients at higher risk of an unsuccessful initiation.

Second, we found that none of the supervised learning algorithms were able to improve prediction about providers' dose decisions based on the clinical information available. In other

**Table 5. Q table.** Expected reward for each action for each cluster. Based on alpha (learning rate) = 0.05 and gamma (discount factor) = 0.2. Both alpha and gamma range from 0 to 1.

| Cluster | Keep Dose | Lower Dose |
|---|---|---|
| **1** | 0.0 | 0.0 |
| **2** | -0.0057 | 0.0 |
| **3** | 0.0 | 0.0 |
| **4** | -0.00002 | 0.0 |
| **5** | -0.227 | -2.26 |
| **6** | -0.021 | 0.0 |
| **7** | 0.0 | 0.0 |
| **8** | -0.00015 | 0.0 |

words, we were unable to 'mimic' the decisions of providers using a statistical model when it came to making dose adjustments of dofetilide. This finding suggests that future efforts based on a gold standard of human decision-making may not lead to the desired outcomes of creating a computer algorithm to replace humans in the process, and that focusing efforts on approaches using reinforcement learning may be a better option.

The key difference of reinforcement learning is that it allows the computer to 'learn' its own approach to obtain a given reward, rather than relying on human behavior as the gold standard. This finding has already been noted in creation of algorithms to win at the board game Go[4, 18], in which the AlphaGo algorithm based on supervised learning of human decisions [18] was bested by the AlphaGoZero algorithm, which learned entirely on its own, without attempting to replicate human decisions[4]. Reinforcement learning has been studied for many years[19, 20], although the medical applications of reinforcement learning are only in their infancy, and there is clearly an opportunity for this approach to greatly improve on clinical decision-making. A number of investigators have recently used this approach to enhance decision-making in clinical care[21], including in the intensive care unit[22].

Interestingly, while use of 8 clusters provided reasonable accuracy (96.1%) with regard to the actual decision made by clinicians, use of smaller numbers of clusters (k = 4 and k = 6) resulted in less accuracy, despite the fact that both of the methods with fewer clusters had more complete Q table (less values of 0.0) and that examination of the first two PCs appeared to suggest that 6 clusters may be a reasonable grouping for the data (S1 Fig). Examination of the characteristics of the clusters for k = 6 (S2 Table) reveals that dose number itself was not the only determinant of cluster composition, as several clusters were composed of mixed dose numbers, although all clusters were composed of sequential dose numbers (for example, no clusters were composed of dose numbers that were out of order, e.g., dose 1 and dose 5). This finding raises a critical issue regarding examination of reinforcement learning for guiding clinical decisions, which is that surrogate outcomes, such as consistency with actual decisions, may not be the ideal approach for identification of the 'optimal' model for guiding decisions to achieve a goal, which in this case was the probability of a successful loading of dofetilide. In that regard, our study highlights a key limitation in applications of machine learning in healthcare data, in which the practical process of data and technology integration limits the ability to build better learning systems. This study was entirely observational, which is in great contrast with most other reinforcement learning applications in which the learning agent is able to practice and improve its policy based on interaction with the environment. A key principle in reinforcement learning is exploration[15], in which better policies can be found by randomly attempting a new action that has been found to already provide the best reward. Without the ability to act on behalf of the policies learned, we were unable to determine if these actions are truly the optimal ones, or if there are conditions in which a decision to change the dose (perhaps at an earlier time in the loading course) could result in a greater likelihood of successful initiation. Whether this limitation was also responsible for the difference in accuracy with use of different cluster numbers, or the lack of convergence we observed using linear function approximation, which has been described in other circumstances[23, 24], remains to be determined. Only through future prospective applications can we verify that the approach applied in this study is the best method to maximize likelihood of successful dofetilide initiation.

## Limitations

There were a number of key limitations in this study. First, we did not examine long-term outcomes, including recurrence of AF or drug toxicity, including *torsade de pointes*. This latter limitation is of obvious importance, as the ultimate goal of the 3-day monitoring period is to

prevent toxicity[11]; however, there are benefits to identification of factors and approaches to maximize safe initiation of dofetilide as we identified, which can lead to improved patient satisfaction and cost savings. A second limitation was that our investigation was limited to the modest number of covariates collected on patients undergoing dofetilide initiation. To truly capture the benefits of many methods of machine learning, particularly deep learning, we would need to have a much larger number of patients and variables to include in the model. In the future, through more efficient data collection and storage, especially of high-density data such as telemetry information, we will be able to further leverage these 'big data' methods to improve healthcare decision-making[25, 26]. Finally, as discussed above, we were unable to prospectively apply and further improve the policy models developed from the observations in this data. Future implementations of these models within a reinforcement learning framework will be needed to determine if this approach is optimal, or if there are better algorithms for ensuring safe and efficient initiation of dofetilide and other medications.

In conclusion, we found that although most patients admitted for initiation of dofetilide are able to successfully complete the loading protocol (i.e., discharged on dofetilide), reinforcement learning approaches to model dose adjustments offer promise to optimize decision making. Future investigations are needed to explore this emerging approach to machine learning and automated clinical decision support.

## Supporting information

**S1 Fig.**
(TIF)

**S2 Fig.**
(TIF)

**S1 Supplemental Methods.**
(DOCX)

**S1 Table.**
(DOCX)

**S2 Table.**
(DOCX)

**S3 Table.**
(DOCX)

## Author Contributions

**Conceptualization:** Andrew E. Levy, Minakshi Biswas, Khaldoun Tarakji, Mina Chung, Peter A. Noseworthy, Christopher Newton-Cheh, Michael A. Rosenberg.

**Data curation:** Khaldoun Tarakji, Mina Chung, Peter A. Noseworthy, Christopher Newton-Cheh, Michael A. Rosenberg.

**Formal analysis:** Rachel Weber, Michael A. Rosenberg.

**Funding acquisition:** Christopher Newton-Cheh, Michael A. Rosenberg.

**Investigation:** Andrew E. Levy, Minakshi Biswas, Khaldoun Tarakji, Peter A. Noseworthy, Christopher Newton-Cheh, Michael A. Rosenberg.

**Methodology:** Andrew E. Levy, Minakshi Biswas, Rachel Weber, Mina Chung, Peter A. Noseworthy, Christopher Newton-Cheh, Michael A. Rosenberg.

**Project administration:** Khaldoun Tarakji, Mina Chung, Peter A. Noseworthy, Christopher Newton-Cheh, Michael A. Rosenberg.

**Resources:** Christopher Newton-Cheh.

**Supervision:** Christopher Newton-Cheh.

**Writing – original draft:** Andrew E. Levy, Michael A. Rosenberg.

**Writing – review & editing:** Andrew E. Levy, Minakshi Biswas, Khaldoun Tarakji, Mina Chung, Peter A. Noseworthy, Christopher Newton-Cheh, Michael A. Rosenberg.

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
