## [Decision Letter · Decision Letter 0]

26 Sep 2019

PONE-D-19-17964

Applications of Machine Learning in Decision Analysis for Dose Management for Dofetilide

PLOS ONE

Dear Dr. Rosenberg,

Thank you for submitting your manuscript to PLOS ONE. After careful consideration, we feel that it has merit but does not fully meet PLOS ONE’s publication criteria as it currently stands. Therefore, we invite you to submit a revised version of the manuscript that addresses the points raised during the review process.As outlined by the Reviewer, the manuscript is interesting in its subject matter, but needs extensive revision for proper evaluation of what the Authors actually did.  It should meet the standard that others in the field should be able to reproduce it and give confidence to those not expert in the field of the methodology used (i.e. why each method was chosen over other possibilities) and what the limitations of the method and interpretation are.

We would appreciate receiving your revised manuscript by Nov 09 2019 11:59PM. To enhance the reproducibility of your results, we recommend that if applicable you deposit your laboratory protocols in protocols.io, where a protocol can be assigned its own identifier (DOI) such that it can be cited independently in the future. For instructions see: http://journals.plos.org/plosone/s/submission-guidelines#loc-laboratory-protocols

We look forward to receiving your revised manuscript.

Kind regards,

Randall Lee Rasmusson

Academic Editor

PLOS ONE

Journal Requirements:

Reviewers' comments:

Reviewer's Responses to Questions

**Comments to the Author**

1. Is the manuscript technically sound, and do the data support the conclusions?

Reviewer #1: Yes

Reviewer #2: Partly

2. Has the statistical analysis been performed appropriately and rigorously? 

Reviewer #1: Yes

Reviewer #2: I Don't Know

3. Have the authors made all data underlying the findings in their manuscript fully available?

Reviewer #1: Yes

Reviewer #2: No

4. Is the manuscript presented in an intelligible fashion and written in standard English?

Reviewer #1: Yes

Reviewer #2: Yes

5. Review Comments to the Author

Reviewer #1: This paper is devoted to an investigation of the interesting problem of the artificial intelligence application for the analysis of clinical data. Specifically, the authors used reinforcement learning algorithm for prediction of successful start of dofetilide loading. They investigated several multiple predictors and multiple supervised approaches. The authors found that the dose adjustment was a significant negative predictor of successful drug initiation.

In my view, the paper is interesting, the authors applied multiple techniques for data analysis, and the obtained results are also interesting and seem solid. I have only minor comments.

Minor points

1. The manuscript needs to have page numbers. So my comments will refer to pages starting from the title page as page #1.

2. Page 3, last paragraph. I would suggest to add that dofetilide is the rapid delayed rectifier current blocker, IKr.

3. Page 3, last paragraph. Two references are incomplete or unclear (#10, #11).

4. Page 5, last paragraph. PC needs to be described as abbreviation somewhere.

5. Page 6, first paragraph. How the rewards -10 and 0 were chosen? Have you investigated other reward values?

6. Page 6, second paragraph from the bottom. The authors examined learning rate from 0.5 to 0.3, but choose alpha = 0.1. Please explain. Or 0.5 should be 0.05?

7. Page 7, first paragraph. Please use other Greek letter other than alpha, as alpha was assigned to the learning rate.

8. Page 7, second paragraph from the bottom. Description of the results in Table 3 is unclear. Give more details. Please also spell out AUC and give definitions of all scores in Table 3.

9. Page 8, first paragraph. Figure 2A does not show any jump when the number of clusters increases. Can the authors explain this feature? Related just below.

10. Page 8, first paragraph. Is there any reason to show 6 clusters in Fig. 2A and 8 clusters in Table 4? Table 4 is not described. Are there significant differences between the clusters in Table 4? Is it possible to plot 8 clusters in Fig. 2b?

11. Page 9, sentence: “Reinforcement learning is only in its infancy in applications outside of computer games”. I disagree with the statement that reinforcement learning is in its infancy. It was known at least 25-30 years ago with corresponding applications. See, for example, JDR Millan and C Torras, A reinforcement connectionist approach to robot path finding in non-maze-like environment, Machine Learning 8: 363-395, 1992; or Gullapalli, Neural Network 3: 671-692, 1990 and references therein.

12. Table 1. Please spell out abbreviations at the bottom of the table 1.

13. Table 5. Title says that “Alpha is sometimes described as the learning rate”. It is actually defined as the learning rate in this manuscript; needs to be proper stated.

Reviewer #2: The problem is an interesting one. However, the description lacks details about the research methodology. For example, the description of the role of the clusters in the reinforcement learning process is unclear as well as the role of PCA. It appears to have only been used to determine the number of clusters, which seems like an odd criteria for selecting a number of clusters. My biggest concern regards the mismatch between the stated outcome of interest, "successful loading of dofetilide", and the reported outcome - percent agreement with physician dosing decisions. I am unable to reconcile this mismatch. It may be that there is great value in the research that was performed. However, the paper, as it currently stands, does not make that value clear to the reader.

6. PLOS authors have the option to publish the peer review history of their article (what does this mean?). If published, this will include your full peer review and any attached files.

Reviewer #1: No

Reviewer #2: No

---

## [Author Response · Author response to Decision Letter 0]

14 Nov 2019

We thank the reviewers for their time and valuable insight. Comments were very helpful in guiding additional work on this investigation, and plans for future studies. Below are the direct responses to the comments, with reference to the changes in the manuscript, as indicated. Comments are listed in bold, with responses given in italics.

Reviewer #1: This paper is devoted to an investigation of the interesting problem of the artificial intelligence application for the analysis of clinical data. Specifically, the authors used reinforcement learning algorithm for prediction of successful start of dofetilide loading. They investigated several multiple predictors and multiple supervised approaches. The authors found that the dose adjustment was a significant negative predictor of successful drug initiation.

In my view, the paper is interesting, the authors applied multiple techniques for data analysis, and the obtained results are also interesting and seem solid. I have only minor comments.

Minor points

1. The manuscript needs to have page numbers. So my comments will refer to pages starting from the title page as page #1.

--We apologize for the exclusion of page numbers, and have added page numbers to the revised manuscript

2. Page 3, last paragraph. I would suggest to add that dofetilide is the rapid delayed rectifier current blocker, IKr.

--We have included this information as suggested on page 3 of the revised manuscript

3. Page 3, last paragraph. Two references are incomplete or unclear (#10, #11).

-- We have relabeled reference 10 to indicate that this is the Tikosyn label prescribing information (published by Pfizer, Inc.), with URL; and we have updated reference 11, which has since been published. 

4. Page 5, last paragraph. PC needs to be described as abbreviation somewhere.

-- We have expanded this abbreviation on page 5 of the revised manuscript as requested.

5. Page 6, first paragraph. How the rewards -10 and 0 were chosen? Have you investigated other reward values?

-- We selected the direction (sign) of the reward based on the notion that the desired outcome was to avoid discontinuation of the medication, and as such assigned a negative reward for a decision (dose adjustment) that led to discontinuation of the loading protocol, with reward of zero for decisions that did not result in immediate discontinuation. This approach has been applied in similar reinforcement learning applications designed to guide decisions away from a negative outcome (e.g., balance a pole in a cart), although we are unaware of well-developed analysis in the medical decision-making literature about a systematic approach to reward selection. Due to the small sample size available for this investigation, we were unable to conduct a formal assessment of different reward values, although in preliminary analyses the absolute value of the reward did not make a qualitative difference in our results (i.e., it did not impact the relative values of the Q table as might have caused the model to select a different action for each cluster). We have included the rationale on page 6 of the revised manuscript.

6. Page 6, second paragraph from the bottom. The authors examined learning rate from 0.5 to 0.3, but choose alpha = 0.1. Please explain. Or 0.5 should be 0.05?

-- The reviewer is correct that this range is 0.05 to 0.3, and that the 0.5 should be 0.05, which has been corrected in the revised manuscript.

7. Page 7, first paragraph. Please use other Greek letter other than alpha, as alpha was assigned to the learning rate.

-- The application of alpha as described on page 7 is the alpha typically applied null-hypothesis testing as the probability of a false-positive result. To avoid confusion, as the reviewer mentions, we have replaced this ‘alpha’ with ‘probability of a false positive’ in the revised manuscript. 

8. Page 7, second paragraph from the bottom. Description of the results in Table 3 is unclear. Give more details. Please also spell out AUC and give definitions of all scores in Table 3.

¬-- We have provided more information on page 7 about the definition and value of the naïve classifier used to compare machine learning approaches, and have defined each of the additional classification metrics in the figure description of Table 3, as requested. The implications of this finding are described in detail in the Discussion section on page 9. 

9. Page 8, first paragraph. Figure 2A does not show any jump when the number of clusters increases. Can the authors explain this feature? Related just below.

-- As highlighted here, and by Reviewer #2 below, the justification for selection of the number of clusters based solely on the principal component analysis was likely insufficient, and required a formal cluster analysis in order to demonstrate that the number of clusters selected captures a reasonable amount of information (sufficiently small intra-cluster variability). In the new Figure 2B, we have provided the cluster analysis, which suggests that a number of clusters of 4 or greater provides adequate clustering of the data (intra-cluster sum-of-squares distance), based on the ‘elbow’ approach. In realization that a smaller number of clusters may provide a simpler interpretation, we repeated the tabular reinforcement learning algorithm using k = 4 and k = 6 clusters, finding that while use of a smaller number of clusters appeared to provide a more complete Q table (fewer zero cells), use of these lower numbers of clusters did not provide the same degree of accuracy with regard to the actual clinical decisions made by providers when examined on the held-out testing set. To provide the reader with additional insight into lower cluster numbers, we have included the cluster descriptions and Q tables for k = 4 and k = 6 clusters in the Supplemental Material section. We discuss the potential explanations and ramifications of this issue below, and in the revised discussion on page 10.

10. Page 8, first paragraph. Is there any reason to show 6 clusters in Fig. 2A and 8 clusters in Table 4? Table 4 is not described. Are there significant differences between the clusters in Table 4? Is it possible to plot 8 clusters in Fig. 2b?

¬-- As above, we revised the cluster selection approach, and selected k = 8 clusters as this number appeared to be most consistent with clinician decisions. The original Figure 2B, which shows the natural clustering across the first two principal components, has been moved to the Supplemental Figures, and replaced with a new Figure 2B, which displays the cluster analysis. 

11. Page 9, sentence: “Reinforcement learning is only in its infancy in applications outside of computer games”. I disagree with the statement that reinforcement learning is in its infancy. It was known at least 25-30 years ago with corresponding applications. See, for example, JDR Millan and C Torras, A reinforcement connectionist approach to robot path finding in non-maze-like environment, Machine Learning 8: 363-395, 1992; or Gullapalli, Neural Network 3: 671-692, 1990 and references therein.

-- We agree with the reviewer, and have changed this paragraph to indicate that it is the medical applications of reinforcement learning that are in their infancy. We have also included the suggested citations (#19 and 20).

12. Table 1. Please spell out abbreviations at the bottom of the table 1.

-- We have added an abbreviations section below Table 1, as well as including units for all measurements provided within Table 1 of the revised manuscript.

13. Table 5. Title says that “Alpha is sometimes described as the learning rate”. It is actually defined as the learning rate in this manuscript; needs to be proper stated.

-- We agree with the reviewer that this information is redundant and have removed it from the Table 5 description in the revised manuscript. 

Reviewer #2: The problem is an interesting one. However, the description lacks details about the research methodology. For example, the description of the role of the clusters in the reinforcement learning process is unclear as well as the role of PCA. It appears to have only been used to determine the number of clusters, which seems like an odd criteria for selecting a number of clusters. 

 -- We completely agree with the reviewer with regard to the manner in which the number of clusters selected was justified based on the PCA, and have included a formal cluster analysis (New Figure 2B), as well as description of additional clustering numbers in the Supplemental Figures (see above response to Reviewer 1 for additional details). As discussed in more detail below, these comments were very insightful for our team as we found a difference in consistency with actual decisions using smaller numbers of clusters. 

My biggest concern regards the mismatch between the stated outcome of interest, "successful loading of dofetilide", and the reported outcome - percent agreement with physician dosing decisions. I am unable to reconcile this mismatch. It may be that there is great value in the research that was performed. However, the paper, as it currently stands, does not make that value clear to the reader.

 -- The reviewer raises a very important point, which we have addressed in the revised discussion on page 10, about the different potential outcomes that can be used to develop an automated (machine-learning) model to guide dosing of dofetilide during loading. The overall goal of this investigation was to identify an approach that would lead to the greatest probability of a successful load, acknowledging that there are indeed safety reasons that certain patients may not be successfully loaded on dofetilide (QT prolongation), but that there may be some room to improve the process of finding the ‘right’ dose for a patient that provides the lowest degree of toxicity while maximizing the clinical efficacy (primarily suppression of atrial fibrillation in this case). 

 In this largely explorative investigation, we examined several machine-learning approaches to improve the probability of a successful load, based on supervised learning (attempting to mimic the decision-making process of clinicians) and reinforcement learning (using feedback in the form of rewards to determine whether to change or keep the same dose at each dosing time point). In order to compare models, we selected the outcome of consistency with the actual decision made by clinicians as a surrogate to the overall goal of a successful load since this single outcome was associated with the highest probability of a successful load in univariate logistic regression models (Table 2). For supervised learning approaches, it was a moot issue as none of the models performed better than a naïve classifier (always keep the same dose) in terms of accuracy and other classification metrics (Table 3). However, for examination of the reinforcement learning models, we noted that use of a smaller number of clusters (k = 4 and k = 6) resulted in less consistency with clinician decisions, to which the reviewer’s point is well-taken that consistency with the actual decision may not indeed be the best for comparing models. Reinforcement learning models are created using rewards assigned according to the overall goal of a successful load, and ultimately, true validation would require prospective application of the model, with dose-decisions based on the model (Q table), which was beyond the scope of our investigation, although planned for future studies.

---

## [Decision Letter · Decision Letter 1]

18 Dec 2019

Applications of Machine Learning in Decision Analysis for Dose Management for Dofetilide

PONE-D-19-17964R1

Dear Dr. Rosenberg,

We are pleased to inform you that your manuscript has been judged scientifically suitable for publication and will be formally accepted for publication once it complies with all outstanding technical requirements.

With kind regards,

Randall Lee Rasmusson

Academic Editor

PLOS ONE

Additional Editor Comments (optional):

Reviewers' comments:

Reviewer's Responses to Questions

**Comments to the Author**

1. If the authors have adequately addressed your comments raised in a previous round of review and you feel that this manuscript is now acceptable for publication, you may indicate that here to bypass the “Comments to the Author” section, enter your conflict of interest statement in the “Confidential to Editor” section, and submit your "Accept" recommendation.

Reviewer #1: All comments have been addressed

2. Is the manuscript technically sound, and do the data support the conclusions?

Reviewer #1: Yes

3. Has the statistical analysis been performed appropriately and rigorously? 

Reviewer #1: Yes

4. Have the authors made all data underlying the findings in their manuscript fully available?

Reviewer #1: Yes

5. Is the manuscript presented in an intelligible fashion and written in standard English?

Reviewer #1: Yes

6. Review Comments to the Author

Reviewer #1: The authors addressed all my comments properly, so I have no further concerns.

7. PLOS authors have the option to publish the peer review history of their article (what does this mean?). If published, this will include your full peer review and any attached files.

Reviewer #1: No

---

## [Editor Report · Acceptance letter]

20 Dec 2019

PONE-D-19-17964R1 

Applications of Machine Learning in Decision Analysis for Dose Management for Dofetilide 

Dear Dr. Rosenberg:

I am pleased to inform you that your manuscript has been deemed suitable for publication in PLOS ONE. Congratulations! Your manuscript is now with our production department. 

With kind regards,

on behalf of

Dr. Randall Lee Rasmusson 

Academic Editor

PLOS ONE